# Preoperative Risk Factors for Conversion from Laparoscopic to Open Cholecystectomy: A Systematic Review and Meta-Analysis

**DOI:** 10.3390/ijerph20010408

**Published:** 2022-12-27

**Authors:** Roberta Magnano San Lio, Martina Barchitta, Andrea Maugeri, Serafino Quartarone, Guido Basile, Antonella Agodi

**Affiliations:** 1Department of Medical and Surgical Sciences and Advanced Technologies “GF Ingrassia”, University of Catania, 95123 Catania, Italy; 2Department of General Surgery and Medical-Surgical Specialties, University of Catania, 95123 Catania, Italy

**Keywords:** conversion, laparoscopic cholecystectomy, open surgery, risk factors

## Abstract

Laparoscopic cholecystectomy is a standard treatment for patients with gallstones in the gallbladder. However, multiple risk factors affect the probability of conversion from laparoscopic cholecystectomy to open surgery. A greater understanding of the preoperative factors related to conversion is crucial to improve patient safety. In the present systematic review, we summarized the current knowledge about the main factors associated with conversion. Next, we carried out several meta-analyses to evaluate the impact of independent clinical risk factors on conversion rate. Male gender (OR = 1.907; 95%CI = 1.254–2.901), age > 60 years (OR = 4.324; 95%CI = 3.396–5.506), acute cholecystitis (OR = 5.475; 95%CI = 2.959–10.130), diabetes (OR = 2.576; 95%CI = 1.687–3.934), hypertension (OR = 1.931; 95%CI = 1.018–3.662), heart diseases (OR = 2.947; 95%CI = 1.047–8.296), obesity (OR = 2.228; 95%CI = 1.162–4.271), and previous upper abdominal surgery (OR = 3.301; 95%CI = 1.965–5.543) increased the probability of conversion. Our analysis of clinical factors suggested the presence of different preoperative conditions, which are non-modifiable but could be useful for planning the surgical scenario and improving the post-operatory phase.

## 1. Introduction

The laparoscopic cholecystectomy (LC)—firstly introduced in the early 1990s—is a routine surgical procedure used for the treatment of gallstones in the gallbladder, usually characterized by cholesterol or bilirubin undissolved in bile [1]. Currently, LC is considered the gold standard, with many advantages over traditional open cholecystectomy (OC), such as related decreased postoperative complications and shorter hospitalization [2]. The consensus for LC, a mini-invasive technique with a great cosmetic result and significant cost reductions, is still growing [2,3,4]. However, OC still has a place in the treatment of gallstones in the gallbladder. Specifically, OC is principally preserved for the challenging cases in which laparoscopy fails [2]. The term “conversion” refers only to cases in which cholecystectomy starts in laparoscopic and finishes in a laparotomic way [5]. Thus, the well-selected cases with specific indications for the open technique (e.g., suspected or confirmed gallbladder cancer) are not considered as cases of conversion [2]. Current literature suggests that the rate of intraoperative conversion from LC to OC ranges from 1 to 15% [6]. Conversion rates vary widely, depending on several risk factors—including old age [7], male gender [8,9], obesity [10,11], previous abdominal surgery [12,13], diabetes [3] and acute cholecystitis [14]. Despite that patient-specific factors associated with a higher likelihood of conversion have been thoroughly studied, previous reviews agreed on only two important risk factors—male gender and old age [15,16]. For this reason, further efforts are needed to avoid the potential complications brought through an intraoperative conversion from LC to OC. The preoperative factors that predict conversion could help to stratify patients and to improve decision making [4,17]. The time elapses to decide whether to change from LC to OC is related to complications during the surgery and higher rates of mortality [1,3,5,17].

A greater understanding of these factors related to conversion is crucial to allow safer procedures and better surgical planning.

In this scenario, the main aim of the present systematic review was to summarize the current knowledge about the main factors associated with conversion from LC to OC in patients with gallstone disease in the gallbladder. Next, we carried out several meta-analyses to evaluate the impact of independent risk factors on conversion rate.

## 2. Materials and Methods

### 2.1. Literature Search

The current systematic review and meta-analysis were conducted in accordance with the Preferred Reporting Items for Systematic Review and Meta-analyses (PRISMA) statements and the Cochrane Handbook’s guidelines (PRISMA checklist available in Appendix A). A systematic literature search in the PubMed, Medline and Web of Science databases was carried out for observational studies investigating risk factors for conversion from laparoscopic cholecystectomy to open surgery in patients with gallstone disease in the gallbladder. Literature search was conducted independently by two authors using the following keywords: (predicting factor OR prediction OR preoperative risk factor) AND (laparoscopic cholecystectomy OR cholecystectomy) AND (gall-bladder OR gallbladder OR gallbladder diseases) AND (conversion to open surgery OR conversion). The databases were searched from September 2021 to May 2022 without language restriction; abstracts and unpublished studies were not included. Moreover, the reference lists from selected articles, including relevant review papers, were searched through to identify all relevant studies. The PRISMA guidelines were followed.

### 2.2. Selection Criteria

Studies were selected only if they satisfied the following criteria: (i) observational studies; (ii) patients with gallstone disease in gallbladder; (iii) treated by laparoscopic cholecystectomy; (iv) consideration of conversion from video-laparoscopic to open cholecystectomy; (v) focus on which factors could increase the probability to have a conversion. Conversely, the following criteria of exclusion were used: (i) studies on patients with other gallstone diseases (e.g., choledocholithiasis); (ii) where cholecystectomy was part of another procedure (e.g., acute cholecystitis, gallbladder cancer, gangrenous cholecystitis); or (iii) where cholecystectomy was carried out exclusively as an emergency treatment; (iv) patients that had indication for cholecystectomy in other principal disease (e.g., symptomatic gallstone disease in cirrhotic patients; cholecystectomy in acute biliary pancreatitis); (v) patients receiving other treatments (e.g., percutaneous cholecystostomy, partial cholecystectomy); (vi) studies recruiting patients < 16 years old; (vi) letters, case reports and case series, clinical trials, previous systematic reviews and meta-analyses, non-English articles and abstracts without full text.

### 2.3. Study Selection and Data Extraction

Two of the authors independently assessed for inclusion all the references identified through the literature search. From all the eligible studies, two authors independently extracted the following information in a standard format: first author’s last name, year of publication, country, period of study (years), study design, number of participants, and conversion rate. Where available, OR and 95%CI of the following risk factors for conversion from LC to OC were also extracted: (i) clinical factors: age, gender, anatomical ambiguity, ASA score, presence of inflammation, admission status, smoking, acute cholecystitis, diabetes mellitus, obesity, hypertension, heart disease, neurological disease, renal disease, lung disease, obstructive jaundice, previous upper abdominal. surgery, and gallstone pancreatitis; (ii) laboratory factors: WBC, AST, ALT, ALP, bilirubin, hemoglobin, LDH, γ-GTP, lipase, albumin, PCR, platelet count, neutrophil ratio; (iii) radiological factors: cholecystitis, gallbladder polyps, gallbladder wall thickness, pericholecystic fluid, impacted stone at the neck, intrahepatic duct dilation, common bile duct diameter, multiple stone, pericholecystic edema. Cross-checked data were entered into STATA software (version 17) by the first author and checked for accuracy by the second author. During study selection and data extraction, any disagreements were resolved by discussion and consensus with a third author.

### 2.4. Risk of Bias and Quality Assessment

The methodological quality of the studies included in the meta-analyses was assessed using the Newcastle–Ottawa scale (NOS), as recommended by the Cochrane Collaboration [18]. The NOS scale rates the studies on selection, comparability, and outcome with a score from 0 to 9. A score of 0 stars indicates the highest possible degree of bias, whereas 9 stars indicate the lowest degree of bias.

### 2.5. Statistical Analysis

Strength of association between each clinical factor and probability of conversion from LC to OC was estimated as ORs (95% CIs). The significance of pooled OR was determined by the Z test. Heterogeneity across studies was measured using the Q test, considering significant statistical heterogeneity as *p* < 0.1. As the Q test only indicates the presence of heterogeneity and not its magnitude, we also reported the I2 statistic, which estimates the percentage of outcome variability that can be attributed to heterogeneity across studies. An I2 value of 0% denotes no observed heterogeneity, whereas 25% is “low”, 50% is “moderate” and 75% is “high” heterogeneity [19]. According to heterogeneity across studies, we used the fixed-effects model (Mantel–Haenszel method) in case of negligible heterogeneity, or the random-effects models (DerSimonian–Laird method) when heterogeneity was significant. All statistical analyses were performed using STATA 17 (STATA Corp, College Station, TX, USA).

## 3. Results

### 3.1. Study Characteristics

The detailed steps of the study selection are given as a PRISMA flow diagram in Figure 1. A total of 427 articles were retrieved from the databases, and 99 duplicates were removed. From 328 articles, 187 were excluded after reading titles and/or abstracts. From these, 53 were excluded according to selection criteria, whereas 35 studies were included in the systematic review.

Table 1 summarizes the main characteristics of observational studies included in the systematic review. In general, studies were conducted between 1990 and 2019 and published from 1997 to 2021. The majority of studies were retrospective [2,3,4,5,17,21,22,23,24,25,26,27,28,29,30,31,32,33,34,35,36,37,38,39,40], while only nine were prospective [1,41,42,43,44,45,46,47,48]. The study conducted by Goonawardena and colleagues reported both retrospective and prospective data [49]. A total of four studies were conducted in Turkey [30,31,37,41], four in in United States [17,33,36,43], three studies in India [45,47,48], two in Japan [21,29], two in Australia [26,49], two in the United Kingdom [40,44], two in Poland [3,42], and two in Ireland [32,35]. Only one study was conducted in Mexico [5], Lebanon [4], Greece [23], Pakistan [25], Egypt [2], Thailand [1], Netherlands [27], China [28], Singapore [34], Iraq [46], Pakistan [38], Saudi-Arabia [39], and Iran [24]. Similarly, only one study was conducted in Italy [22]. Overall, sample sizes ranged from 73 to 11,669, with a mean age from 35 to 68.35. Instead, the percentage of men ranged from 23.3 to 95.1%.

All studies included in the systematic review reported conversion rate from laparoscopy to open surgery during cholecystectomy. Single values from all studies ranged from 1% to 23.3%, while the estimated pooled proportion of conversion rate was 5.99% (95%CI = 4.92–7.16%; Figure 2).

### 3.2. Risk Factors for Conversion from Laparoscopy to Open Surgery during Cholecystectomy

Studies included in the systematic review also evaluated the main risk factors associated with the probability of conversion from LC to OC. Increasing age has been reported as a risk factor for conversion by the difficult dissection secondary to a repeated number of attacks of inflammation and ensuing thickening and fibrosis of the gallbladder [3,22,23,25,28]. In addition, male gender has been considered as a risk factor for conversion, probably due to the increased severity of gallstone disease and the more frequent adhesions and inflammations observed in men [4,17,31]. However, results reported by the included studies were controversial. In particular, only the study conducted by Kaafarani et al. estimated that, among the population of Veteran American soldiers, conversion rate was 95.1% in male participants [42]. Van der Velden and colleagues, instead, did not report the exact proportion of men and women with cholecystectomy conversion [27].

Beyond age and gender, additional risk factors (e.g., clinical, laboratory and radiological factors) could lead to changes from laparoscopy to open surgery during cholecystectomy. For this reason, the studies included in the systematic review investigated the impact of these risk factors—individually or as a cumulative effect—on conversion rate.

All patients included in the systematic review received a diagnosis of gallstones in the gallbladder by imaging and were subjected to planned laparoscopic cholecystectomy. Among the clinical factors, admission status, acute cholecystitis, diabetes, obesity, hypertension, heart disease and previous abdominal surgery were frequently investigated. With respect to the admission status, patients treated in emergency showed higher probability of conversion than those with elective treatment [3,4,5,24,27,28,31,34,43,44,49]. Similarly, acute cholecystitis—defined as onset due to cholecystolithiasis just before surgery—was considered as a risk factor for conversion [1,2,3,17,21,22,23,25,27,29,31,33,34,35,37,44,47,49]. In this context, previous comorbidities (i.e., hypertension, diabetes, obesity and heart diseases) [2,3,4,5,17,23,30,31,39,43,46,47,49], cirrhosis [2,34,38] and upper abdominal surgery [2,4,23,24,26,28,30,33,35,37,41,43,46] also increased the probability of conversion during the cholecystectomy due to physiological and anatomical difficulties. Three studies considered neurological diseases as risk factor for conversion [3,4,43], with Al Masri and colleagues also assessing renal disease [4].

Moreover, six studies identified the preoperative endoscopic retrograde cholangial-pancreatography (ERCP) as a significant factor to predict conversion [2,4,30,31,33,44]. The obstructive jaundice has been found in some studies as a factor related to conversion, although it is mainly due to the association with a form of cholangitis [2,41,49]. Pancreatitis is reported in three studies [23,44,46].

Regarding the main laboratory tests, some authors did not report any parameters. On the other hand, several studies evaluated the main values that could be associated with conversion, even if the evidence is still controversial. For instance, white blood cell (WBC) count, albumin, and glomerular filtration rate were significantly associated with conversion in several studies [17,24,25,28,30,33,34,37,41,43,49]. The liver function test (LFT) with the measurement of AST [22,30,43], ALT [5,17,22,30,34], ALP [4,17,22,24,30,33,34,37,39,49], GGT [4,21,22], bilirubin total [4,17,24,30,33,39,49] and direct [4,21,39] were the most frequent flags measured in these patients. Few studies reported a significant association of conversion rate with fibrinogen [22], platelet count and INR [43]. Moreover, three studies considered the C-reactive protein as significative as the risk factors for conversion [21,22,28].

Some studies investigated the impact of gallbladder (GB) thickness > 4 mm in conversion [1,2,17,24,25,26,27,28,29,30,32,33,34,38,40,41,44,45,47,48,49], while Raman et al. analyzed the different sizes of thickness in relation to conversion rate [36]. It has been well-established that the analysis of the various dimensions of GB at ultrasound is crucial during the dissection of GB. For this reason, a methodical approach is required: (i) the measure of GB major axis [4,5,27] and (ii) the echogenicity of the GB’s content, in which it is possible to find a single stone or multiple stones [2,27,30,47], but also biliary sand and biliary sludge. Moreover, the presence of stones in the Hourtman’s pouch—corresponding to the neck of the gallbladder—and the contracted GB were also considered as significant elements involved in conversion [1,32]. Dilated common bile duct (CBD) and choledocholithiasis were also considered in several studies [2,4,5,17,21,24,25,26,27,28,30,32,33,35,41,44,45,46,49], as well as the intra-hepatic ducts (IHDs), because the ultrasound diagnostic method leads to assessment of the real state of the intra- and extra-hepatic bile ducts for planning the most appropriate procedure [5,49]. In this context, US Murphy Sign [25,28,37,49] and hepatomegaly [5] were factors significantly related with conversion.

With respect to operative findings, anatomical ambiguity was considered a risk factor for higher probability of conversion [2,3,4,5,17,21,22,23,25,27,28,29,30,31,34,35,36,37,41,42,46,47,49]. In addition, inflamed GB (i.e., complication of acute cholecystitis) [3,17,21,22,23,25,30,31,34,39,42], peritoneal adhesion [2,5,17,21,22,25,27,28,29,30,31,32,33,34,36,38,39,45,46] and fistula of GB (i.e., consequences of chronic cholecystitis) [2,22,23,30,31,33,34,36,42,46,49] were indicated as risk factors for conversion. During laparoscopic cholecystectomy, some studies reported that bleeding [2,4,5,21,22,23,27,29,30,31,32,33,36,39,41,42,44,45,46,47,49], bile duct injury [2,5,17,21,22,23,25,28,29,30,31,32,33,41,42,44,46,47] and bowel injury [31,36,41,44,46] were the main complications leading to conversion. Moreover, grade of difficultly [44], inability to create pneumoperitoneum [2,23,34,36] and anesthesia complications [2] were considered as risk factors for conversion. Mirizzi’s Syndrome and cancer of gallbladder were significantly associated with conversion in several studies [5,22,23,33,34,38,45], but their diagnosis was accidental and occurred during GB extraction.

### 3.3. The Association of Clinical Factors with the Probability of Conversion

Among studies included in the systematic review, 11 with available data on the association between clinical factors and the risk of conversion were included in the meta-analyses (Table 2). We performed individual meta-analyses for the following clinical factors analyzed by at least three studies: gender, age, acute cholecystitis, admission status, diabetes, hypertension, heart diseases, obesity, and previous upper abdominal surgery.

Overall, 10 studies estimated the probability of conversion associated with male gender. The Q test and I^2^ statistics showed significant heterogeneity between studies (*p* < 0.1; I^2^ = 85.9%). Using the random effect model, the meta-analysis showed that male gender increased the probability of conversion (OR = 1.907; 95%CI = 1.254–2.901; Figure 3A).

Overall, five studies estimated the probability of conversion associated with age over 60 years. The Q test and I^2^ statistics showed no significant heterogeneity across studies (*p* > 0.1; I^2^ = 0.0%). Using the fixed effect model, the meta-analysis showed that age over 60 years increased the probability of conversion (OR = 4.324; 95%CI = 3.396–5.506; Figure 3B).

Overall, five studies estimated the probability of conversion associated with admission status. The Q test and I^2^ statistics showed significant heterogeneity across studies (*p* < 0.1; I^2^ = 93.9%). Using the random effect model, however, no association was evident (OR = 2.350; 95%CI = 0.991–5.574; Figure 3C).

Overall, six studies estimated the probability of conversion associated with acute cholecystitis. The Q test and I^2^ statistics showed significant heterogeneity across studies (*p* < 0.1; I^2^ = 84.4%). Using the random effect model, the meta-analysis showed that acute cholecystitis increased the probability of conversion (OR = 5.475; 95%CI = 2.959–10.130; Figure 4A).

Overall, three studies estimated the probability of conversion associated with heart diseases. The Q test and I^2^ statistics showed significant heterogeneity across studies (*p* < 0.1; I^2^ = 74.1%). The pooled OR, obtained through the random effect model, was 2.947 (95%CI = 1.047–8.296), showing that heart diseases increased the probability of conversion (Figure 4B).

Overall, seven studies estimated the probability of conversion associated with previous upper abdominal surgery. The Q test and I^2^ statistics showed significant heterogeneity across studies (*p* < 0.1; I^2^ = 77.3%). The pooled OR, obtained through the random effect model, was 3.301 (95%CI = 1.965–5.543), showing that previous upper abdominal surgery increased the probability of conversion (Figure 4C).

Overall, seven studies estimated the probability of conversion associated with diabetes. The Q test and I^2^ statistics showed significant heterogeneity across studies (*p* < 0.1; I^2^ = 73.2%). Using the random effect model, the meta-analysis showed that diabetes increased the probability of conversion (OR = 2.576; 95%CI = 1.687–3.934; Figure 5A).

Overall, five studies estimated the probability of conversion associated with hypertension. The Q test and I^2^ statistics showed significant heterogeneity across studies (*p* < 0.1; I^2^ = 90%). Using the random effect model, the meta-analysis showed that hypertension increased the probability of conversion (OR = 1.931; 95%CI = 1.018–3.662; Figure 5B).

Overall, five studies estimated the probability of conversion associated with obesity (BMI ≥ 30 Kg/m^2^). The Q test and I^2^ statistics showed significant heterogeneity across studies (*p* < 0.1; I^2^ = 84.8%). The pooled OR, obtained through the random effect model, was 2.228 (95%CI = 1.162–4.271), showing that obesity increased the probability of conversion (Figure 5C).

### 3.4. Quality of Studies

As reported, the methodological quality of the studies included in the meta-analyses were assessed using the Newcastle–Ottawa scale (NOS). The quality of case-control studies ranged from seven to nine stars, with the definition of cases and ascertainment of exposure being the items most affecting the quality of studies. With respect to prospective studies, the quality assessed was eight stars, with the representativeness of the exposed cohort as the item most affecting the quality of studies.

## 4. Discussion

In the present systematic review and meta-analysis, we aimed to identify the main preoperative patient-related risk factors for conversion from LC to OC in patients with gallstone disease in the gallbladder. The analyses of these factors could help to redefine the surgical strategy, improving patient safety. In some cases, conversion may be necessary to prevent injury and treat intraoperative complications [3]. Overall, the studies included have investigated the association of several risk factors—male sex, older age, BMI > 30 kg/m^2^, thickness of gallbladder wall > 3 mm, previous abdominal surgery and comorbidities—with the risk of conversion. However, findings were controversial, and the available data were heterogeneous, making it difficult to summarize the evidence in a systematic way. Moreover, not all studies have analyzed the same risk factors affecting conversion. For this reason, we carried out a systematic review of 35 studies to estimate conversion rate and, after, to investigate the main risk factors for conversion. Although conversion rate reported by each study ranged from 1% to 23.3%, our results showed a pooled conversion rate of 6.0%.

Among studies included in the systematic review, 11 with available data on the association between clinical factors and the risk of conversion were included in individual meta-analyses. Firstly, 10 studies evaluated the relationship between male gender and the probability of conversion from LC to OC [3,4,17,21,23,24,25,28,43,49]. Several studies reported male gender as a risk factor for conversion, with ORs ranging from 1.6 to 5.9 [3,4,17,23,28,43,49]. However, there were also studies that did not demonstrate any association between male gender and conversion [21,24,25]. When we pooled all studies together, we noted 1.91 increased odds of conversion in male patients than in females. The underlying reason is unclear, but it may be attributable to body fat distribution, which is different in males and females, making the laparoscopic approach generally more difficult in men than women. Moreover, it has been proposed that men are less likely than women to seek medical advises [50,51]. Inflammation and dense adhesions were also frequently cited as reasons for conversion in men [17]. Among the main characteristics, five studies evaluated the relationship between age and the probability of conversion from LC to OC [3,22,23,25,28]. All studies demonstrated the positive association between age over 60 years and conversion, with ORs ranging from 3.0 to 6.2. This finding was confirmed by our meta-analysis, showing that age over 60 years increased the probability of conversion, with an OR equal to 4.3. It is plausible that older people have a longer history of gallstones and an increased number of cholecystitis attacks [52,53,54].

Acute cholecystitis can also affect conversion rate, as investigated by six studies included in our meta-analysis [3,22,23,25,28,49]. Only one study did not find any statistical association between acute cholecystitis and conversion [23]. When we pooled all studies together, we noted 5.5 increased odds of conversion in patients with acute cholecystitis versus in their counterpart. The high rates of conversion from LC to OC for acute cholecystitis could depend on a technical difficulty of managing severe inflammatory adhesions around the inflamed gallbladder, making the dissection of Calot’s triangle and the recognition of the anatomy more difficult.

In this context, the comorbidities such as diabetes and hypertension were associated with a higher conversion rate, probably because both complications could determine a higher risk of infections and a general poor condition of the patient. Seven studies evaluated the relationship between diabetes and the probability of conversion from LC to OC [3,4,17,23,25,28,49]. Several studies reported diabetes as a risk factor for conversion, with ORs ranging from 1.9 to 5.5 [3,4,17,23,49]. However, there were also studies that did not demonstrate any association between diabetes and conversion [25,28]. When we pooled all studies together, we demonstrated that diabetes increased the probability of conversion, with an OR equal to 2.6, confirming that such a relationship is due to the presence of acute inflammation or changes in the wall from microvascular diseases. Moreover, some diabetic patients may not develop symptoms of gallbladder disease early due to autonomic and peripheral neuropathy, leading to delayed diagnosis and a greater risk for conversion. Similarly, three studies suggested the association between hypertension and higher probability of conversion from LC to OC, with ORs ranging from 1.2 to 7.0 [3,4,43]. This finding was confirmed by our meta-analysis, reporting an OR equal to 1.9.

Among coexisting diseases in patients undergoing cholecystectomy; in addition, obesity and heart diseases could be considered as risk factors for conversion from LC to OC. Obesity has been previously described as an important factor for conversion, probably for causes related to excessive intra-abdominal fat, difficult mobilization of the liver, placing trocars and moving the instruments due to the thickness of the abdominal wall [5]. Previous studies investigated the relationship between obesity and conversion; however, the results were controversial probably due to few obese patients [4,23,25,28,49]. Our meta-analysis showed 2.2 increased odds of conversion in diabetic patients than in their counterpart. Similarly, when we pooled all studies investigating the role of heart diseases as risk factors for conversion [3,4,23], we demonstrated 2.9 increased probability of conversion for patients with previous heart diseases.

In this scenario, previous abdominal surgery has been considered as a risk factor for conversion, due to existing peritoneal adhesions, which make it difficult to perform gallbladder dissection. Overall, seven studies estimated the probability of conversion associated with previous upper abdominal surgery, with ORs ranging from 1.6 to 15.5 [4,23,24,25,28,43,49]. After applying the meta-analysis, the pooled OR was 3.3, confirming that a history of upper abdominal surgery increased the probability of conversion.

Overall, our findings confirmed what was evident in a previous meta-analysis, where the preoperative predictive factors reported were age > 65 years, male gender, acute cholecystitis, history of diabetes, thickened gallbladder walls and previous upper abdominal surgery [55]. In addition, the meta-analysis by Rothman and colleagues demonstrated that gallbladder wall > 4–5 mm, age> 60 years, male gender, and acute cholecystitis were risk factors for conversion from LC to OC [16]. The analysis of clinical factors suggested the presence of different preoperative variables, which are non-modifiable but could be useful for planning the surgical scenario and improving the post-operatory phase. Our systematic review and meta-analysis provided robust implications for systematic quality improvement efforts. The present analysis could improve clinical practice, promoting appropriate patient counseling throughout accurate risk factor assessment. A better understanding of preoperative risk factors for conversion from LC to OC might help prevent conversion by optimizing patient clinical condition. Alternatively, the preoperative optimization of patients with comorbidities could set a more realistic evaluation of conversion risk and subsequent recovery time. For this reason, it should be necessary to improve the management of these patients before surgery. A key strategy may be represented by the validation of specific recommendations for clinicians and their subsequent training. Next, ad hoc surveys could be developed for evaluating the adherence of clinicians to the guidelines for the management of these patients. In this scenario, it is well known that radiological and laboratory factors should also be studied for their potential impact on the risk of conversion from LC to OC.

Our work had some limitations to be considered. Firstly, for the large period considered, the technologies and knowledge regarding laparoscopic cholecystectomy suffered great changes during the years, as well as conversion rate, which did not always remain the same. Included studies were heterogeneous in terms of study period. We recognize that excluding studies conducted before a certain year or applying a meta-regression to evaluate the impact of study period could be important. However, the majority of studies were conducted during a wide range of time. For this reason, we cannot apply subgroup analyses and/or meta-regressions. Another limitation is the retrospective nature of the data collection, which may lead to a limited ability to correctly classify the preoperative diagnosis and make the analyzed group very heterogeneous. Although we excluded all epidemiological studies in which cholecystectomy was carried out exclusively as an emergency treatment, the meta-analysis included studies in which patients were treated in elective surgery and/or both in elective and in emergency surgery. To evaluate the impact of admission status on conversion rate, we performed a specific meta-analysis that, however, did not point out a significant association. Moreover, the lack of data (e.g., ORs and 95%CI) did not allow us to carry out meta-analyses for other important risk factors for conversion (e.g., radiological and laboratory factors).

## 5. Conclusions

In conclusion, the present systematic review and meta-analysis identified the most frequent factors associated with conversion from LC to OC. Our results demonstrated that an accurately derived estimation of the risk for conversion could be obtained from readily available preoperative data. However, further studies with larger sample size, more accurate information, and standardized definitions and protocols are needed.

## Figures and Tables

**Figure 1 ijerph-20-00408-f001:**
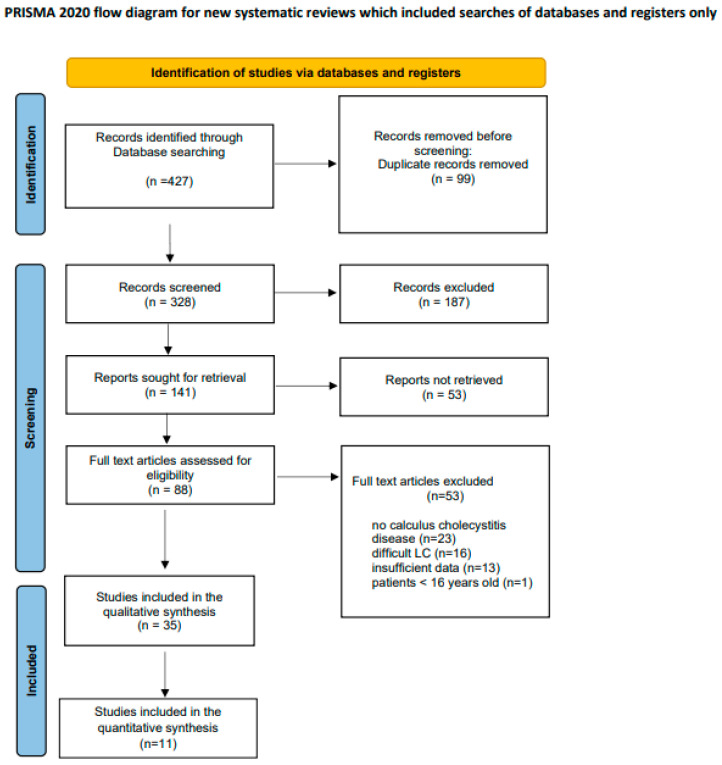
PRISMA flow diagram of study selection [20].

**Figure 2 ijerph-20-00408-f002:**
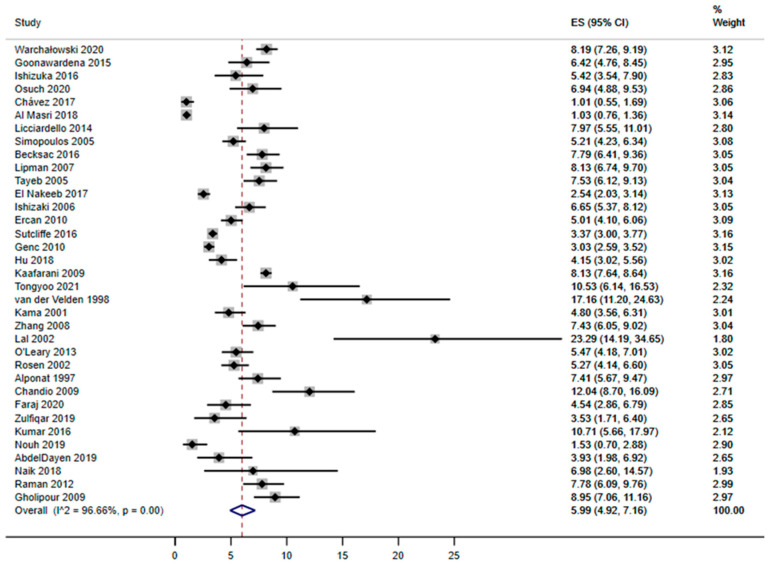
Forest plot of the pooled conversion rate [1,2,3,4,5,17,21,22,23,24,25,26,27,28,29,30,31,32,33,34,35,36,37,38,39,40,41,42,43,44,45,46,47,48,49]. The graph shows individual and pooled conversion rates (expressed in terms of percentage).

**Figure 3 ijerph-20-00408-f003:**
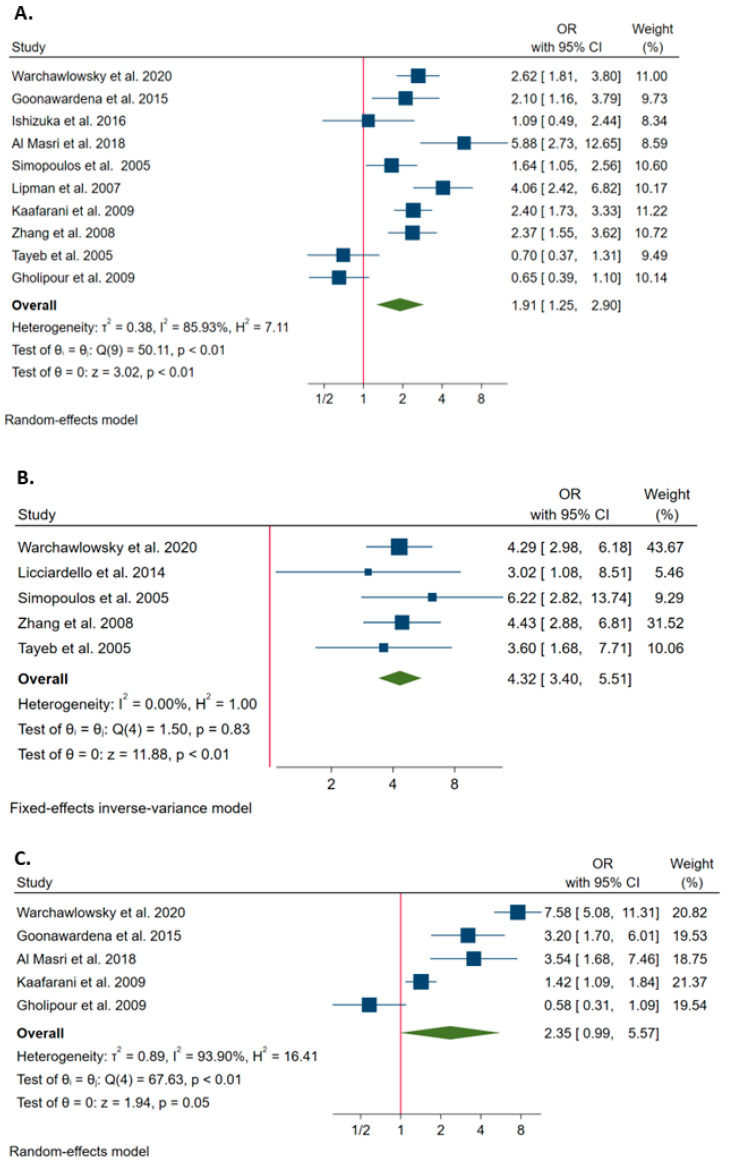
Forest plots of the associations of male gender (**A**) [3,4,17,21,23,24,25,28,43,49], age over 60 years (**B**) [3,22,23,25,28] and admission status (**C**) [3,4,24,43,49] with the probability of conversion. The graph (**A**) shows the ORs for the comparison between male and female genders (reference group). The graph (**B**) shows the ORs for the comparison between age > 60 years and ≤ 60 years (reference group). The graph (**C**) shows the ORs for the comparison between emergency and elective treatment (reference group).

**Figure 4 ijerph-20-00408-f004:**
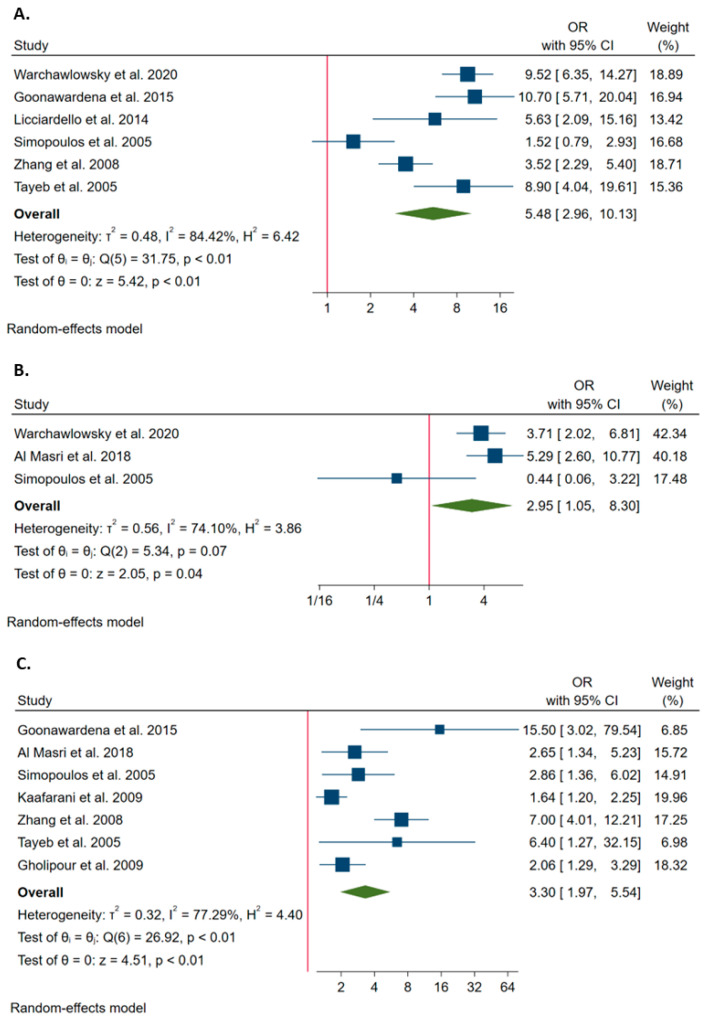
Forest plots of the associations of acute cholecystitis (**A**) [3,22,23,25,28,49], heart disease (**B**) [3,4,23] and previous upper abdominal surgery (**C**) [4,23,24,25,28,43,49] with the probability of conversion. The graph (**A**) shows the ORs for the comparison between acute and non-acute cholecystitis (reference group). The graph (**B**) shows the ORs for the comparison between patients with heart disease and those without it (reference group). The graph (**C**) shows the ORs for the comparison between patients with previous upper abdominal surgery and those without it (reference group).

**Figure 5 ijerph-20-00408-f005:**
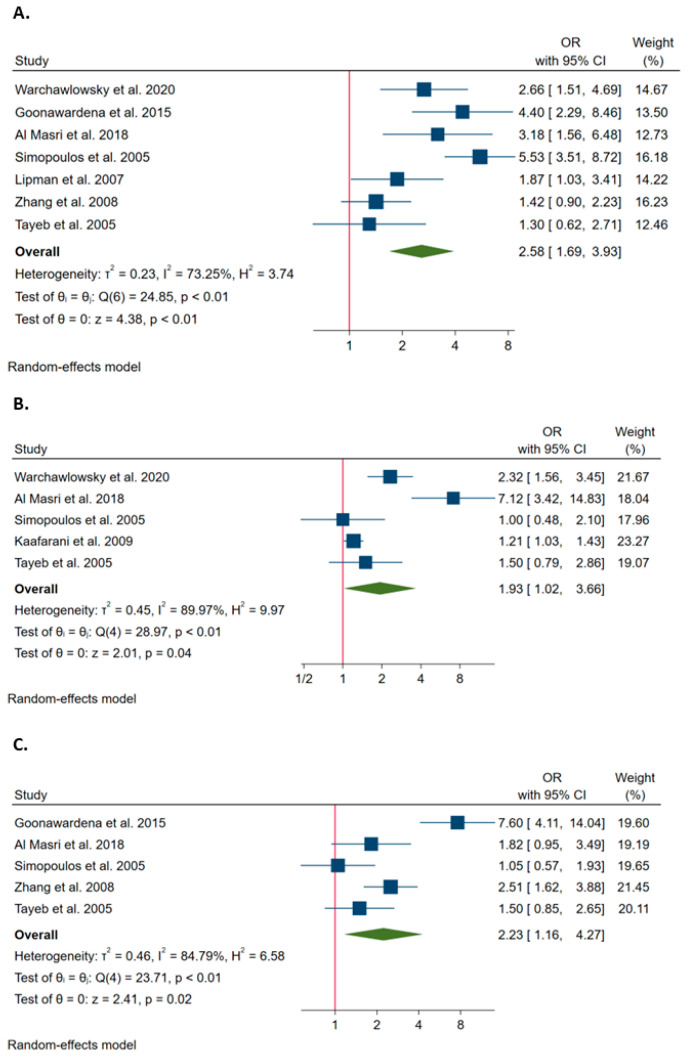
Forest plots of the associations of diabetes (**A**) [3,4,17,23,25,28,49], hypertension (**B**) [3,4,23,25,43] and obesity (**C**) [4,23,25,28,49] with the probability of conversion. The graph (**A**) shows the ORs for the comparison between patients with diabetes and those without it (reference group). The graph (**B**) shows the ORs for the comparison between patients with hypertension and those without it (reference group). The graph (**C**) shows the ORs for the comparison between obese and non-obese patients (reference group).

**Table 1 ijerph-20-00408-t001:** Characteristics of studies included in the systematic review [1,2,3,4,5,17,21,22,23,24,25,26,27,28,29,30,31,32,33,34,35,36,37,38,39,40,41,42,43,44,45,46,47,48,49].

First Author and Publication Year	Country	Study Period	Study Design	Number of Patients	Age (Mean and SD)	Male (%)	Conversion Rate (%)
Warchałowski et al., 2020 [3]	Poland	2008–2018	Retrospective	3213	65.7 ± 1.4	120 (45.6)	263 (8.2)
Goonawardena et al., 2015 [49]	Australia	2010–2012	Prospective and retrospective	732	56 ± 18	20(43)	47 (6.4)
Ishizuka et al., 2016 [21]	Japan	2000–2010	Retrospective	461	NA	231 (50.1)	25 (5.4)
Osuch et al., 2020 [42]	Poland	2013–2018	Prospective	504	54.8 ± 16.9	22 (62.9)	35 (6.9)
Chávez et al., 2017 [5]	Mexico	2009–2013	Retrospective	1386	42 ± 15	5 (35.7)	14 (1)
Al Masri et al., 2018 [4]	Lebanon	2000–2015	Retrospective	4668	68.35 ± 13.7	37 (80.4)	48 (1.03)
Licciardello et al., 2014 [22]	Italy	2011–2013	Retrospective	414	65.9 ± 13.9	14 (42.5)	33 (7.9)
Simopoulos et al., 2005 [23]	Greece	1992–2004	Retrospective	1804	52.66 ± 14.7	31 (33.0)	94 (5.2)
Beksac et al., 2016 [37]	Turkey	2006–2011	Retrospective	1335	61.9 ± 12.5	57 (54.8)	104 (7.8)
Lipman et al., 2007 [17]	Ohio (USA)	2000–2005	Retrospective	1377	NA	57 (50.9)	112 (8.1)
Tayeb et al., 2005 [25]	Pakistan	1997–2001	Retrospective	1249	44.2 ± 12.4	17 (23.3)	94 (23.3)
El Nakeeb et al., 2017 [2]	Egypt	2011–2016	Retrospective	3269	54	47 (56.6)	83 (2.5)
Ishizaki et al., 2006 [29]	Japan	1993–2004	Retrospective	1339	58 ± 13	57 (64)	89 (7.5)
Ercan et al., 2010 [30]	Turkey	2002–2007	Retrospective	2015	57.8 ± 13.0	60 (59.4)	101 (5.0)
Sutcliffe et al., 2016 [44]	UK	2014	Prospective	8820	51 ± 17	110 (49.8)	297 (3.4)
Genc et al., 2010 [31]	Turkey	1999–2010	Retrospective	5382	49.34 ± 9.9	84 (51.5)	163 (3.2)
Hu et al., 2018 [26]	Australia	2013–2016	Retrospective	1035	61 ± 15	24 (55.8)	43 (4.2)
Kaafarani et al., 2009 [43]	USA	2005–2008	Prospective	11,669	63.9 ± 12.4	903 (95.1)	949 (8.1)
Tongyoo et al., 2021 [1]	Thailand	2018–2019	Prospective	152	57.61 ± 13.8	9 (56.2)	16 (10.5)
Van der Velden et al., 1998 [27]	Netherlands	1993–1996	Retrospective	134	50.7	NA	23 (17.2)
Kama et al., 2001 [41]	Turkey	1992–1999	Prospective	1000	43.8	22 (45.8)	48 (4.8)
Zhang et al., 2008 [28]	China	2005–2006	Retrospective	1265	60.0 ± 11.9	50 (53.2)	94 (7.4)
Lal et al., 2002 [45]	India	1999–2000	Prospective	73	35	NA	17 (23.3)
O’Leary et al., 2013 [32]	Ireland	2000–2006	Retrospective	1061	43.3	31 (53.4)	58 (5.5)
Rosen et al., 2002 [33]	USA	1996–2000	Retrospective	1347	59.9	29 (41)	71 (5.3)
Alponat et al., 1997 [34]	Singapore	1990–1995	Retrospective	783	NA	22 (38)	58 (7.4)
Chandio et al., 2009 [35]	Ireland	2004–2006	Retrospective	324	61	NA	39 (12)
Faraj et al., 2020 [46]	Iraq	2019	Prospective	485	52 ± 17.3	11 (50)	22 (4.5)
Zulfiqar et al., 2019 [38]	Pakistan	2017–2018	Retrospective	283	NA	NA	10 (3.5)
Kumar et al., 2016 [47]	India	2013–2015	Prospective	112	43.4	5 (41.7)	12 (10.7)
Nouh et al., 2019 [39]	Saudi Arabia	2014–2015	Retrospective	589	NA	2 (32.2)	9 (1.5)
AbdelDayen et al., 2019 [40]	UK	2013–2014	Retrospective	280	49.9	NA	11 (3.9)
Naik et al., 2018 [48]	India	2016–2017	Prospective	86	48.92 ± 11.9	2 (33.3)	6 (6.9)
Raman et al., 2012 [36]	USA	2006–2009	Retrospective	874	NA	NA	68 (7.8)
Gholipour et al., 2009 [24]	Iran	1997–2004	Retrospective	793	48.9 ± 15	NA	71 (9)

**Table 2 ijerph-20-00408-t002:** Risk factors examined by studies included in the meta-analyses [3,4,17,21,22,23,24,25,28,43,49].

First Author and Publication Year	Risk Factors for Conversion
Warchałowski et al., 2020 [3]	Gender, Age, Anatomical ambiguity, Admission Status, Acute Cholecystitis, Choledocholithiasis, Diabetes Mellitus, Hypertension, Heart Disease, Neurological Disease
Goonawardena et al., 2015 [49]	Gender, Age, Ethnicity, ASA > III, Acute Cholecystitis, Admission Status, BMI > 30, Obstructive Jaundice, Previous Upper Abdominal Surgery, Gallstone pancreatitis, Diabetes Mellitus, WBC, ALP, Bilirubin, Lipase, PCR, Choledocholithiasis, Cholecystitis, Gallbladder Wall width, Pericholecystic fluid, Impacted stone at the neck, Intrahepatic Duct dilatation
Ishizuka et al., 2016 [21]	Gender, Obesity, WBC, AST, ALT, ALP, Bilirubin, LDH, γ-GT, Albumin, PCR, Platelet count, Neutrophil ratio
Al Masri et al., 2018 [4]	Gender, Age, Admission Status, Obesity, Smoke, Previous Upper Abdominal Surgery, Diabetes Mellitus, Hypertension, Heart Disease, Dyslipidemia, Neurological Disease, Renal Disease, Lung Disease, WBC, AST, ALT, ALP, Bilirubin, Hemoglobin, γ-GT, Lipase, Platelet count, Choledocholithiasis, Gallbladder Wall width, Pericholecystic fluid, Impacted stone at the neck
Licciardello et al., 2014 [22]	Age, Acute Cholecystitis
Simopoulos et al., 2005 [23]	Gender, Age, Acute Cholecystitis, Presence of inflammation, Obesity, Previous Upper Abdominal Surgery, Diabetes Mellitus, Hypertension, Heart Disease
Lipman et al., 2007 [17]	Gender, Diabetes Mellitus, WBC, Bilirubin, Albumin, Pericholecystic fluid
Tayeb et al., 2005 [25]	Age, Acute Cholecystitis, Obesity, Obstructive jaundice, Previous Upper Abdominal Surgery, Gallstone pancreatitis, Tenderness in RUQ, Palpable Gallbladder, Diabetes Mellitus, Hypertension, WBC, ALP, Bilirubin, Gallbladder Wall with, Pericholecystic fluid, Dilatation of CBD, Multiple stones, US-Murphy sign
Kaafarani et al., 2009 [43]	Gender, Admission status, Previous Upper Abdominal Surgery, Hypertension, WBC, Albumin
Zhang et al., 2008 [28]	Gender, Age, Acute Cholecystitis, Previous Upper Abdominal Surgery, Diabetes Mellitus, Obesity, WBC, Gallbladder Wall width
Gholipour et al., 2009 [24]	Gender, Age, Admission Status, Smoke, Previous Upper Abdominal Surgery, WBC, ALP, Bilirubin, Gallbladder Wall width, Dilatation of CBD, Pericholecystic edema

## Data Availability

Not applicable.

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
