# Peer review of "Preoperative Risk Factors for Conversion from Laparoscopic to Open Cholecystectomy: A Systematic Review and Meta-Analysis"

_ijerph, 2022, doi:10.3390/ijerph20010408_

Round 1

Reviewer 1 Report

The fundamental topic and format of this paper are sound. The paper addresses an important topic to evaluate the risk factors which affect the probability of conversion from laparoscopic cholecystectomy to open surgery.

Author Response

Dear Editor,

Please consider the revised version of the manuscript entitled “Preoperative risk factors for the Conversion from Laparoscopic to Open Cholecystectomy: a systematic review and meta-analysis” in which we have considered all comments and suggestions from reviewers. This letter is intended for the convenience of the reviewers and contains the list of the requested changes. The following list of changes and answers to comments of Reviewer addresses all revisions made in the manuscript (in red font).

Reviewer 1

Comment: The fundamental topic and format of this paper are sound. The paper addresses an important topic to evaluate the risk factors which affect the probability of conversion from laparoscopic cholecystectomy to open surgery.

Answer: We are grateful to Reviewer 1 for his/her comments.

Reviewer 2 Report

Paper discusses the risk factors for conversion of lap chole, using systematic review, and meta analysis.

Overall the population looked at is confused.  The literature search needed to include gallbladder AND cholecystectomy - some papers may only talk about cholecystectomy so some key papers may have been missed.

In terms of the inclusion/exclusion criteria, it stated that only elective operations were included.  I know for certain Sutcliffe et all including emergency presentations as well. I'm not sure why only elective procedures were selected as a criteria, and if this is an important factor of the study, then I think it needs to be included in the title, and policed more thoroughly in the paper selection; my preference would be to include all operations, elective and emergency. 

Some of the phrasing is confusion - ie - "In particular, only the study conducted by Kaafarani 163 et al. estimated that 95.1% of male undergone the conversion, probably due to the study 164 population of Veteran American soldiers [43].' This seems to imply 95.1% of men underwent conversion?

Some of the Forrest plots are not clearly labelled, and it is not clear what they represent (even with scrolling to the legend at the bottom)

Author Response

Dear Editor,

Please consider the revised version of the manuscript entitled “Preoperative risk factors for the Conversion from Laparoscopic to Open Cholecystectomy: a systematic review and meta-analysis” in which we have considered all comments and suggestions from reviewers. This letter is intended for the convenience of the reviewers and contains the list of the requested changes. The following list of changes and answers to comments of Reviewer addresses all revisions made in the manuscript (in red font).

Reviewer 2

C: Paper discusses the risk factors for conversion of lap chole, using systematic review, and meta-analysis. Overall, the population looked at is confused. The literature search needed to include gallbladder AND cholecystectomy - some papers may only talk about cholecystectomy so some key papers may have been missed.

A: We are grateful to Reviewer 1 for his/her comments that helped us in improving our manuscript. We apologize if the population looked at is confused. As described in our manuscript, the literature search was conducted using the following keywords: (predicting factor OR prediction OR preoperative risk factor) AND (laparoscopic cholecystectomy OR cholecystectomy) AND (gall-bladder OR gallbladder OR gallbladder diseases) AND (conversion to open surgery OR conversion). Thus, we included observational studies on patients with gallstones disease in gallbladder, treated in by laparoscopic cholecystectomy.

C: In terms of the inclusion/exclusion criteria, it stated that only elective operations were included.  I know for certain Sutcliffe et all including emergency presentations as well. I'm not sure why only elective procedures were selected as a criteria, and if this is an important factor of the study, then I think it needs to be included in the title, and policed more thoroughly in the paper selection; my preference would be to include all operations, elective and emergency.

A: We apologize if the inclusion/exclusion criteria are confused. In our systematic review we excluded all epidemiological studies in which cholecystectomy was carried out exclusively as an emergency treatment. By contrast, we included the studies in which patients were treated in elective surgery and/or both in elective and in emergency surgery. In the latter case, previous studies demonstrated that patients treated in emergency showed higher probability of conversion than those with elective treatment. In our meta-analysis, we included 5 studies with data (i.e., OR and 95%CI) on the association between admission status and the risk of conversion. However, we did not find any significant association. Please, consider changes in the revised version of our manuscript

C: Some of the phrasing is confusion - ie - "In particular, only the study conducted by Kaafarani et al. estimated that 95.1% of male undergone the conversion, probably due to the study population of Veteran American soldiers [43].' This seems to imply 95.1% of men underwent conversion?

A: Thank to Reviewer 2 for his/her comment. Please consider the revised version of our manuscript.

C: Some of the Forrest plots are not clearly labelled, and it is not clear what they represent (even with scrolling to the legend at the bottom)

A: We apologize if forest plots were not so clear. Please consider the changes in the figure legends.

Reviewer 3 Report

Thank you for the possibility to review the article entitled “Preoperative risk factors for the Conversion from Laparoscopic to Open Cholecystectomy: a systematic review and meta-analysis”. The authors did a huge work for this article. I have some remarks.

-        The authors selected only patients scheduled for elective surgery (inclusion criteria), however, many studies reporting data on cholecystitis were included. Please explain.

-        The quantitative synthesis suffers from the inclusion of studies with old data. Patients operated in ’90 are expected to have different conversion rates and outcomes than patients operated in recent years. You could solve the problem by performing a subanalysis for “old” and “new” data.

-        Line 320: The authors stated that the reason why male gender is associated to conversion to open surgery is that males seek medical advises later than female. I don’t think this statement is supported by any evidence. More likely, body fat distribution is different in male and female, and the laparoscopic approach is generally more difficult in men than women. Many articles dealt with this topic in bariatric surgery.

-        Finally, I would add a paragraph in the discussion to explain how the main findings of this article should be applied in clinical practice and what should we change in the management of patients scheduled for laparoscopic cholecystectomy.

Author Response

Dear Editor,

Please consider the revised version of the manuscript entitled “Preoperative risk factors for the Conversion from Laparoscopic to Open Cholecystectomy: a systematic review and meta-analysis” in which we have considered all comments and suggestions from reviewers. This letter is intended for the convenience of the reviewers and contains the list of the requested changes. The following list of changes and answers to comments of Reviewer addresses all revisions made in the manuscript (in red font).

Reviewer 3

C: Thank you for the possibility to review the article entitled “Preoperative risk factors for the Conversion from Laparoscopic to Open Cholecystectomy: a systematic review and meta-analysis”. The authors did a huge work for this article. I have some remarks.

A: We are grateful to Reviewer 3 for his/her comments that helped us in improving our manuscript.

C: The authors selected only patients scheduled for elective surgery (inclusion criteria), however, many studies reporting data on cholecystitis were included. Please explain.

A: We apologize if this point was not so clear. Please consider changes in the description of selection criteria in which we have better explained that we excluded all epidemiological studies in which cholecystectomy was carried out exclusively as an emergency treatment. This choice was done because emergency treatment is usually more prone to conversion. However, in the current systematic review we collected information about admission status (elective vs. emergency treatment) and acute cholecystitis – defined as onset due to cholecystolithiasis just before the surgery - treated in emergency. We included studies in which patients were treated in elective surgery and/or both in elective and in emergency surgery. In our meta-analysis, we included 5 studies with data (i.e., OR and 95%CI) on the association between admission status and the risk of conversion. However, we did not find any significant association. Moreover, in our meta-analysis we included 6 studies estimated the probability of conversion associated with acute cholecystitis, showing that acute cholecystitis increased the probability of conversion. Please consider changes in the revised version of our manuscript

C: The quantitative synthesis suffers from the inclusion of studies with old data. Patients operated in ’90 are expected to have different conversion rates and outcomes than patients operated in recent years. You could solve the problem by performing a subanalysis for “old” and “new” data.

A: As already described in the previous version of our manuscript, there are some limitations to be considered, including the heterogeneity in which the studies were conducted. The first limitation is the large period considered, because the technologies and knowledge regarding laparoscopic cholecystectomy have suffered great changes during the years, as well as the conversion rate, which did not always remain the same. We recognize that excluding studies conducted before a certain year or applying a meta-regression to evaluate the impact of study period could be important. However, the majority of studies were conducted during a wide range of time. For this reason, we cannot apply subgroup analyses and/or meta-regressions. Anyway, we better explained this point in the limitation section.

C: Line 320: The authors stated that the reason why male gender is associated to conversion to open surgery is that males seek medical advises later than female. I don’t think this statement is supported by any evidence. More likely, body fat distribution is different in male and female, and the laparoscopic approach is generally more difficult in men than women. Many articles dealt with this topic in bariatric surgery.

A: We thank the Reviewer for the suggestion. Please consider the revised version of our manuscript

C: Finally, I would add a paragraph in the discussion to explain how the main findings of this article should be applied in clinical practice and what should we change in the management of patients scheduled for laparoscopic cholecystectomy.

A: We thank the Reviewer for the suggestion. Please, consider the revised version of our manuscript

Round 2

Reviewer 3 Report

Authors improved the manuscript and acknowledged study limitations correctly. I have no further comment.